# Solution Combustion Synthesis of Ni-Based Nanocatalyst Using Ethylenediaminetetraacetic Acid and Nickel-Carbon Nanotube Growth Behavior

**DOI:** 10.3390/ma16227191

**Published:** 2023-11-16

**Authors:** Juyoung Kim, Hwanseok Lee, Jaekwang Lee, Hyunjo Yoo, Ilguk Jo, Heesoo Lee

**Affiliations:** 1School of Convergence Science, Pusan National University, Busan 46241, Republic of Korea; juyoungkim@pusan.ac.kr (J.K.); hyunjoyoo@pusan.ac.kr (H.Y.); 2School of Materials Science and Engineering, Pusan National University, Busan 46241, Republic of Korea; hwanseok@pusan.ac.kr (H.L.); jaekwanglee@pusan.ac.kr (J.L.); 3Department of Advanced Materials Engineering, Dong-Eui University, Busan 47340, Republic of Korea

**Keywords:** solution combustion synthesis, chelating agent, fuel-to-oxidizer ratio, nanocatalyst, nanopowder, carbon nanotube, CNT growth

## Abstract

We studied the influence of the ethylenediaminetetraacetic acid (EDTA) content used as combustion fuel when fabricating nickel oxide (NiO) nanocatalysts via solution combustion synthesis, as well as the growth behavior of carbon nanotubes (CNTs) using this catalyst. Nickel nitrate hexahydrate (Ni(NO_3_)_2_∙6H_2_O) was used as the metal precursor (an oxidizer), and the catalysts were synthesized by adjusting the molar ratio of fuel (EDTA) to oxidizer in the range of 1:0.25 to 2.0. The results of the crystal structure analysis showed that as the EDTA content increased beyond the chemical stoichiometric balance with Ni(NO_3_)_2_∙6H_2_O (F/O = 0.25), the proportion of Ni metal within the catalyst particles decreased, and only single-phase NiO was observed. Among the synthesized catalysts, the smallest crystallite size was observed with a 1:1 ratio of Ni ions to EDTA. However, an increase in the amount of EDTA resulted in excessive fuel supply, leading to an increase in crystallite size. Microstructure analysis revealed porous NiO agglomerates due to the use of EDTA, and differences in particle growth based on the fuel ratio were observed. We analyzed the growth behavior of CNTs grown using NiO nanocatalysts through catalytic chemical vapor deposition (CCVD). As the F/O ratio increased, it was observed that the catalyst particles grew excessively beyond hundreds of nanometers, preventing further CNT growth and leading to a rapid termination of CNT growth. Raman spectroscopy was used to analyze the structural characteristics of CNTs, and it was found that the I_D_/I_G_ ratio indicated the highest CNT crystallinity near an F/O ratio of 1:1.

## 1. Introduction

The emergence of the Fourth Industrial Revolution has driven the development of advanced materials to support cutting-edge technologies in extreme environmental conditions. It has also led to the creation of ultra-small electronic materials for applications in fuel cells and the field of biotechnology, expanding beyond the realms explored by humankind so far [1,2,3]. Carbon nanotubes (CNTs) have garnered significant attention as a nanomaterial exhibiting exceptional mechanical, electrical, thermal, and chemical stability. They are considered prime candidates to fulfill the requirements of diverse future technologies [4,5,6,7]. CNTs exhibit outstanding properties not only as standalone materials but also as composites when combined with conventional materials like metals, ceramics, and polymers. This versatility opens up numerous possibilities, including their use as electrical conductors, thermal conductors, electrochemical electrodes, and protective coatings. Research is actively ongoing across various domains encompassing structural materials, energy materials, composite material synthesis, and conductive materials [8,9,10,11].

Multiple methods have been introduced for the production of CNTs such as arc discharge, laser ablation, flame synthesis, high pressure carbon monoxide, and electrolysis [12,13,14,15], with catalytic chemical vapor deposition (CCVD) being one of them. In CCVD, CNTs are grown from the surface of a metal catalyst through a heat treatment in an atmosphere that contains carbon sources such as CO or CH_4_. This method enables the growth of carbon nanotubes (CNTs) using a variety of substrates or catalysts and can be used with any form of carbon source, including solid, liquid, or gaseous forms. Pre-treatment of the catalyst or the substrate on which the CNTs are grown can result in the formation of distinct types of CNTs, such as aligned, straight, and customized architectures. Thus, the preparation of the metal nanocatalyst plays a crucial role because the properties of CNTs depend significantly on catalyst morphology. This method also ensures the proper dispersion of CNT particles in composite materials, a critical challenge in composite material manufacturing. The aforementioned synthesis techniques are simpler compared to other methods; therefore, this method is cost-effective and scalable for the mass production of CNTs [12].

Solution combustion synthesis (SCS) [16,17,18,19,20] is widely employed due to its simple single-step procedure and its effectiveness in synthesizing nanopowders. This technique relies on the exothermic reaction between a metal nitrate precursor and fuel [21,22]. The combustion reactions typically occur on hot plates or in electric furnaces at temperatures below 200–300 °C. The chemical energy released during the exothermic reaction between the precursor and the fuel rapidly heats the system to high temperatures, eliminating the need for further external heating. This self-combustion process generates a significant amount of gas, ultimately increasing the reaction surface area and resulting in the production of a fine powder [20,23]. Consequently, the powders synthesized through combustion exhibit enhanced homogeneity, higher purity, and fewer hard aggregates compared to the powders prepared using conventional solid-state methods [19,20,24]. The choice of metal ions is the first consideration in the research for a nanocatalyst for CNT growth using SCS. The various metals that act as catalysts exhibit different interactions with carbon, as well as have distinct solubilities of carbon. These factors have consequences on the dissolution and precipitation of carbon by metal particles, which can cause deformation and defects in the carbon structure. Furthermore, they can lead to the formation of a core-shell structure of carbon or metal particles [25]. Reactive transition metals such as Ni, Co, and Fe are considered candidates due to their high solubility of carbon at higher temperatures and high melting points, with numerous attempts to control their properties, whether used individually or in combination [26,27,28]. Nickel-based catalysts are widely acknowledged for their exceptional ability to produce CNTs, especially due to their excellent proficiency in cleaving C–C and C–H bonds [29]. These metal ions are introduced in the form of metal nitrates, metal sulfates, or similar compounds and are decomposed in the SCS reaction, with the metal ions becoming the nucleus from which the catalyst grows, with the salts acting as oxidants [12].

The typical fuels employed in SCS reactions include ethylenediaminetetraacetic acid (EDTA), glycine, and urea, and the combination process varies by type, significantly influencing the particle shape of the synthesized nanocatalyst. Furthermore, the fuel-to-oxidizer (F/O) ratio is a critical factor since reactivity is influenced by this ratio, affecting the reaction temperature and rate [30,31,32]. Therefore, the type of metal used, the choice of fuel, and the F/O ratio all have an impact on the size, shape, sintering, structure, and particle composition of the synthesized nanopowder in combustion synthesis.

This study examined the impact of the fuel-to-oxidizer ratio of EDTA and NiO on the synthesis of metal nanocatalysts through SCS, and CNTs were grown using the synthesized catalyst with the CCVD method. Crystalline property and microstructure analyses were conducted to evaluate the phase, shape, and size of the catalyst particles. The shape and crystallinity of the resulting CNTs were investigated through FE-TEM and Raman spectroscopy. These were used to explore how the characteristics of the nanocatalyst particles influence CNT growth when employing the synthesized metal nanocatalyst. 

## 2. Experimental Procedures

The metal nanocatalyst for CNT growth was fabricated using solution combustion synthesis, which is one of the liquid-state synthesis methods using sol-gel. Nickel nitrate hexahydrate (Ni(NO_3_)_2_∙(H_2_O)_6_, Sigma-Aldrich, St. Louis, MO, USA) served as a precursor and an oxidizer for the synthesis. Ethylenediaminetetraacetic acid (C_10_H_16_N_2_O_8_, EDTA, 99%, Alfa aesar, Haverhill, MA, USA) functioned as a chelating agent and fuel for the process of the self-combustion reaction. The required quantity of metal oxidizer was measured and dissolved in a beaker containing a minimum amount of distilled water using a hot plate at 80 °C with magnetic stirring. Subsequently, EDTA was added to the beaker, and the pH was adjusted to 10 using an ammonia solution (NH_4_∙OH, 28%, Junsei, Tokyo, Japan). At this point, the solution was stirred for 4 h at 70 °C using a hot plate to induce a chelating reaction, which led to the formation of a gelled solution. A viscous liquid was obtained after thermally dehydrating the solution. The dried gel was sampled for analysis, and thermogravimetric analysis (TG–DTA, L81–11, Linseis, Selb, Germany) was conducted to determine the temperature of weight reduction, setting the heat treatment temperature for the gel accordingly.

To investigate the influence of the fuel-to-oxidizer (F/O) ratio on powder characteristics, various F/O ratios were chosen. The stoichiometric representation of combustion reactions is typically expressed in terms of an equivalence ratio (*Ø*). Here, *Ø*s/*Ø*m is the stoichiometric F/O ratio derived from propellant chemistry calculations, while *Ø*m (mixture ratio) denotes the F/O ratio effectively employed in combustion reactions. An equivalence ratio of 1 symbolizes a stoichiometrically balanced combustion reaction, where the oxidation state of metal nitrate reduces the valence of the fuel. A positive deviation (*Ø* > 1) implies a fuel-lean composition, while a negative deviation (*Ø* < 1) suggests a fuel-rich composition [33]. Thermodynamic calculations theoretically indicate that the reaction between nickel nitrate hexahydrate and EDTA is as described in Equation (1) below.
$$Ni{\left({NO}_{3}\right)}_{2}\cdot 6H_{2}O+\emptyset \cdot C_{10}H_{16}N_{2}O_{8}+\left(10\emptyset -\frac{5}{2}\right){\cdot O}_{2}$$
(1)$$NiO+10\emptyset \cdot CO_{2}+{\left(8\emptyset +6\right)\cdot H}_{2}O+(\emptyset +1)\cdot N_{2}$$

In this reaction, *Ø* = 1/4 indicates that the initial mixture does not need atmospheric oxygen for the complete oxidation of the fuel, termed as stoichiometric combustion, whereas *Ø* > 1/4 (*Ø* < 1/4) suggests fuel-rich (-lean) conditions. The literature suggests that in fuel-lean conditions, there is inadequate fuel to oxidize the metal ion; hence, the ratio of oxidizer (metal ion) to fuel (EDTA) was 1:0.25 for the stoichiometric balanced condition, and fuel-rich conditions of 1: 0.50, 0.75, 1.00, 1.50, 2.00 provided sufficient energy for the Ni^2+^ to react in actual reactions. The combustion process was initiated by elevating the temperature to 260 °C in an air atmosphere using an electric furnace (CRF M20P, Winner Technology Co., Ltd., Gyeonggi-do, Korea), inducing a self-combustion reaction. The resulting sample was ground and calcined at 600 °C using an electric furnace to obtain NiO nanopowder. The synthesized NiO nanopowder was analyzed using the X-ray diffraction method (XRD; Rigaku, Ultima-IV, Tokyo, Japan) to determine its phase and crystallite size. The morphology was examined using field-emission scanning electron microscopes (FE-SEM; MIRA3, TESCAN, Brno, Czech Republic) at the Converging Materials Core Facility, and energy-dispersive X-ray spectroscopy (EDS; MIRA3, TESCAN) was employed to observe the surface composition of each particle.

The synthesized NiO nanopowder with different F/O ratios was used as a catalyst for CNT growth. CNT growth was carried out using the CCVD method in a methane atmosphere using a box-type electric furnace (M03-0012, Nabertherm, New Castle, DE, USA). The temperature was maintained at 1000 °C for 4 h, and then it was cooled in the open air to ensure complete catalyst reaction and CNT growth. The CNT powder was studied through transmission electron microscopy (FE-TEM, JEOL Ltd., JEM-2100, Tokyo, Japan) to verify the growth behavior from Ni metal ions, and the crystallinity of the Ni nanoparticles was analyzed simultaneously. Additionally, the structure of the grown CNTs was analyzed using Raman spectroscopy (RAMANtouch, Nanophoton, Osaka, Japan).

## 3. Results and Discussion

### 3.1. Thermal Analysis

To determine the temperature at which the thermal decomposition of EDTA begins and ends, the TGA of the sample with the most fuel-rich condition, where the ratio of metal ions to EDTA was 1:2, was measured. The results showed a sharp weight loss due to an exothermic reaction starting around 200 °C, indicating the occurrence of a combustion reaction, as shown in Figure 1a. The weight reduction of EDTA persisted up to around 260 °C. As seen in Figure 1b, this aligns closely with the temperatures of 473 K (200 °C) and 533 K (260 °C), at which the decomposition reactions of EDTA take place. At this point, the decomposed CH_2_COO- moiety encounters Ni^2+^ and forms a metal-EDTA linkage, leading to an amorphous state. This will be explained later with the morphology analysis results. Thus, the production of nanocatalysts involved heat treatment at temperatures of 260 °C, which marked the complete termination of the SCS reaction, and 600 °C, resulting in the decomposition of Ni(NO_3_)_2_∙(H_2_O)_6_ and the formation of NiO. This is illustrated in the heat treatment profile in Figure 1c.

### 3.2. Synthesis of NiO Nanocatalyst

A phase analysis was conducted on the NiO nanocatalyst synthesized through SCS, as depicted in Figure 2. The peaks obtained from the XRD measurements were analyzed using Highscore plus software 3.0c (PANalytical). In all samples, the peaks corresponding to NiO (JCPDS No. 98-009-2132) were identified, confirming the successful synthesis of the intended NiO nanocatalyst. Additionally, a Ni phase (JCPDS No. 98-026-0169) was present, and an observed decrease of the Ni peak aligned with an increase in the percentage of EDTA added as fuel. Theoretically, in samples with an EDTA stoichiometric ratio of 1:0.25, Ni should react sufficiently with the given temperature and fuel to form NiO [33]. However, in actual reactions, the temperature of the self-combustion flame does not reach the theoretical temperature due to experimental losses, which are caused by elements such as C, H, and N contained in the added EDTA reacting with the oxygen, which should react with Ni, resulting in vaporization [32]. The trend in XRD analysis supports this, revealing that as the F/O ratio increases and the amount of added EDTA grows, the presence of more fuel leads to an increased combustion temperature. This facilitates the complete oxidation of Ni, resulting in an increase in the NiO peak and a decrease in the Ni peak.

The crystallite size of the synthesized NiO nanocatalyst was calculated using the Scherrer equation, based on the information obtained from the measured XRD peaks (refer to Figure 3). For stoichiometric conditions at *Ø* = 1, i.e., F/O ratio = 0.25, it was observed that the crystallite size decreased up to the condition where F/O ratio = 1, making *Ø* > 1. An increase in the F/O ratio indicates that a fuel-rich state persists for a longer duration during the combustion reaction. This suggests that as the combustion temperature rises, the powder synthesis concludes rapidly, resulting in synthesis without any crystallite growth. On the other hand, in conditions where *Ø* >> 1, particularly when the F/O ratio exceeds 1.5, it is expected that the fuel continues to ignite even after the reaction ends. This sustained ignition generates energy, which in turn promotes further crystal growth [22]. 

The morphology of the synthesized NiO nanocatalyst was observed using SEM. Figure 4a–c are the SEM images with a magnification of 10,000×, and their enlarged images are shown in Figure 4d–f with magnification of 50,000×. The results revealed agglomerates with an overall weak porous structure. Each of these agglomerates was composed of primary particles ranging from a few nanometers to several hundred nanometers, and it was observed that the shape and size of these primary particles varied with changes in the F/O ratio. The porous structure of the synthesized NiO nanocatalyst is formed due to the significant amount of gas generated during the combustion reaction. As mentioned earlier, in the initial stages of the SCS reaction, external heat is applied to cause the decomposition of the fuel, and a substantial amount of gas is naturally generated during this process. These gases then undergo self-combustion at high temperatures, resulting in the formation of porous structures with nano-sized particles [18,19,20,32].

Figure 4 reveals that in the case (e) with an F/O ratio of 1.00, primary particles ranging from a few to tens of nanometers can be observed as intended. This suggests that the SCS reaction was appropriately terminated, in line with the earlier XRD and crystallite size analysis results. However, in the cases (a) and (d) with an F/O ratio of 0.25, the agglomerate size is small, but the actual particle size shows a larger distribution ranging from a few nanometers to several hundred nanometers. This is likely due to the SCS reaction terminating prematurely due to fuel shortage, resulting in an uneven reaction. For the most fuel-rich condition with an F/O ratio of 2.00 (see Figure 4c,f), although the structure was porous, the agglomerated particle size was large, and the resulting powder had grown with sharper angles, with sizes exceeding several hundred nanometers. This is consistent with previous results, suggesting that the excessive fuel continued to react, leading to an overgrowth of particles.

Figure 5 shows the EDS results for the synthesized NiO nanocatalyst at minimum and maximum F/O ratios of (a) 0.25 and (b) 2.0, respectively. As shown, Ni and O exhibit evenly distributed results under both conditions, and a peak of the same ratio is shown in the component analysis results. A peak of the carbon element was observed, possibly due to the use of carbon tape during the measurement, when compared with the phase analysis results presented in Figure 2. 

### 3.3. Carbon Nanotube Growth via CCVD Method

The synthesized NiO nanocatalysts were used to grow CNTs using the CCVD method. The morphology and crystallinity of the CNTs were analyzed using TEM and Raman spectroscopy. In all conditions, the growth of the CNTs from NiO particles was confirmed through TEM. As previously observed in Figure 4, in Figure 6a, there are minimally agglomerated, relatively straightforward, and thin CNTs compared to those in Figure 6c. For the F/O ratio of 1.00, as seen in Figure 6b, an increased level of agglomeration was observed. In Figure 6a,b, the diameters of the grown CNTs are approximately 150 to 210 nm, and CNTs longer than 1 micrometer can also be observed. Along with thin, elongated CNTs like in (a), the growth of clustered shell-type carbon particles were also evident. Especially in (c), while some growth of CNTs similar to (a) or (b) was identified, most particles showed significant clustering. As seen in the figure, it appears that carbon particles are enveloping the NiO particles. This is thought to be due to the excessive combustion reactions occurring in a fuel-rich state, causing overgrown NiO particles. In this state, the carbon ions obtained from the incoming gas source are unable to grow into CNTs and instead adsorb on the NiO surface, forming a shell structure, as shown in Figure 6c [35].

The grown CNTs were analyzed using their Raman spectrum (see Figure 7). The D-peak and G-peak, which allow for the understanding of the structural properties of CNTs, were magnified and displayed. In the Raman spectrum, the D band (disorder band) and the G band (graphite band) provide crucial information for evaluating the crystallinity of CNTs. The D band indicates structural irregularities like defects or low crystallinity in the carbon structures and corresponds to the vibrations of sp^3^-bonded carbon atoms. A higher intensity suggests the presence of more defects or irregularities in the CNT. The G band corresponds to the vibrations of sp^2^-bonded carbon atoms and represents the carbon structure, with higher intensity implying better crystallinity. By comparing the intensity of the D-peak and G-peak, the crystallinity of nanocarbon structures such as CNTs can be inferred [36]. As shown in Figure 7, as the F/O ratio increased to 1.00, the previously lower G-peak intensity increased, and the D-peak intensity decreased, leading to a decrease in the I_D_/I_G_ ratio. However, when the F/O ratio exceeded 1.00, the I_D_/I_G_ ratio increased again. The variation in peak intensity is similar to the observed CNT growth behavior in Figure 6. As seen in Figure 6c, under conditions of a high fuel ratio, the synthesized NiO nanocatalyst exhibits overgrown catalyst particles in the hundreds of nanometers in size, with the carbon ions forming a shell-like configuration around the catalyst particles. Typically, single-wall CNTs are known to grow on catalysts with sizes of a few nanometers or even less, and even for multi-wall CNTs, the size is typically in the tens of nanometers [15,27]. Carbon ions agglomerated in this way often exceed a size of several hundred nanometers, and lacking the required crystallinity for CNT growth, they assume a state similar to amorphous carbon, resulting in an increase in the D-peak. Through the preceding analyses, it was evident that the morphology of the generated nanocatalyst significantly varied with the F/O ratio. This variation had a substantial impact on CNT growth behavior through CCVD. Therefore, selecting the appropriate fuel ratio is crucial for more effective nanocatalyst synthesis.

## 4. Conclusions

We synthesized a NiO nanocatalyst through SCS, a convenient method for producing nanometal oxides, and observed its behavior in growing CNTs, using it as a catalyst through the CCVD method. Initially, to understand the effect of the fuel on the properties of the nanocatalysts produced through SCS, we varied the F/O ratio from the stoichiometric number of 0.25 to the fuel-rich state of 2.0 using EDTA. Through XRD, we confirmed the creation of the desired nano-sized NiO catalyst without impurities in states that were more fuel-rich than the theoretical stoichiometric proportion. This is thought to counteract the losses that occur in actual experiments, contrary to theory. Similarly, we could obtain powders with the smallest crystallite size of 55.27 nm and spherical shape in fuel-rich situations where *Ø* > 1 compared to the theoretical ratio where *Ø* is 1. In cases where *Ø* >> 1, even after the synthesis of the NiO nanocatalyst was completed, we observed an overgrown crystallite size of 62.87 nm due to the ongoing combustion caused by the remaining fuel. The analysis of the microstructure of the synthesized NiO nanocatalyst confirmed a similar trend. In the case of a fuel-to-oxidizer (F/O) ratio of 0.25, spherical-shaped particles were observed, but their sizes varied from sub-nanometers to hundreds of nanometers in an uneven manner. As the ratio increased to 1.00, the smallest primary particles with spherical shapes could be observed, but with further increases in the amount of added fuel, the particle sizes grew into angular forms measuring up to several hundred nanometers. Additionally, a weak porous agglomerate structure, a byproduct of the SCS reaction, was observed under all conditions. As the fuel ratio increased, the agglomerates became more strongly bonded and grew in size. The altered morphology of the catalyst directly influenced the growth of the carbon nanotubes (CNTs). In situations with a low F/O ratio of 0.25, crystalline CNTs were observed to grow from individual NiO nanocatalyst particles. However, in situations with an F/O ratio of two, where overgrown catalyst particles were present and carbon was adsorbed around them in a shell-like form, the CNTs did not grow and exhibited amorphous characteristics. This was confirmed through Raman spectroscopy, with the highest crystallinity (*I*_D_/*I*_G_ = 0.26) observed at an F/O ratio of 1.00, and a decrease in crystallinity as the fuel ratio increased to 2.00 ((*I*_D_/*I*_G_ = 0.96). Through this research, we were able to identify the significant impact of fuel-ratio selection on the morphology of nanometal catalysts in the SCS process. Furthermore, we analyzed the effect of catalyst morphology on CNT growth using the CCVD method. We plan to continue researching other factors influencing CNT growth behavior and aim to contribute to the effective production and performance enhancement of CNT/ceramic composites based on this foundation.

## Figures and Tables

**Figure 1 materials-16-07191-f001:**
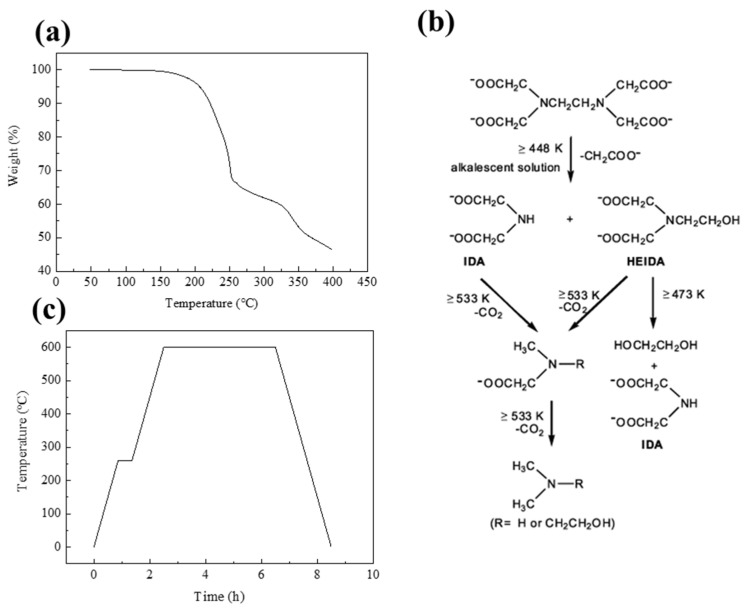
(**a**) TGA of sample with F/O ratio of 2.0, (**b**) the decomposition routes of EDTA [34], and (**c**) temperature profile of calcination during SCS.

**Figure 2 materials-16-07191-f002:**
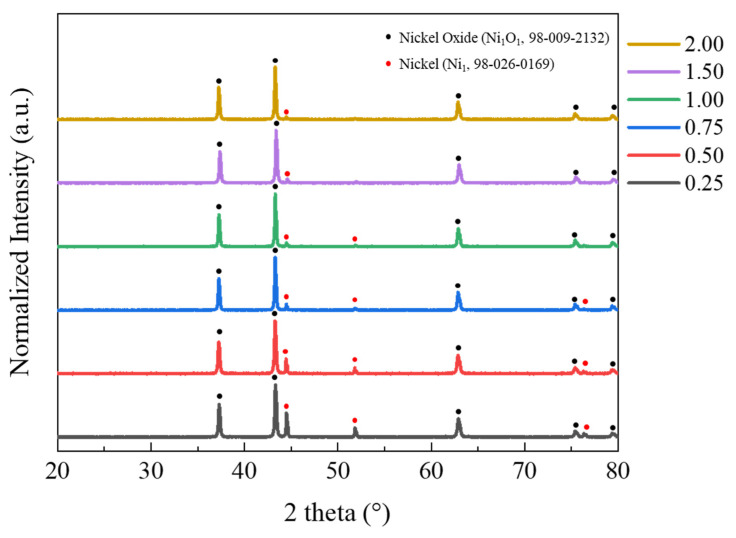
XRD patterns of combustion powders with different F/O ratios.

**Figure 3 materials-16-07191-f003:**
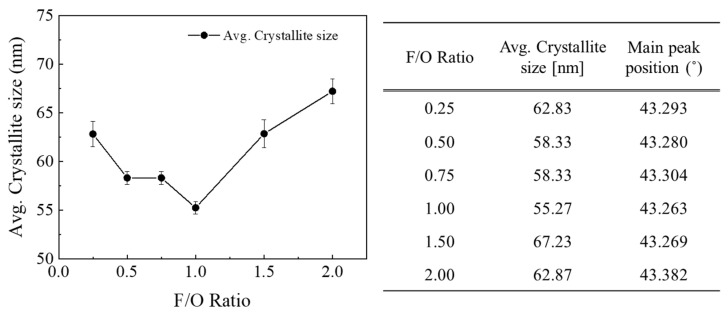
Crystallite size calculated using Scherrer equation with different F/O ratios.

**Figure 4 materials-16-07191-f004:**
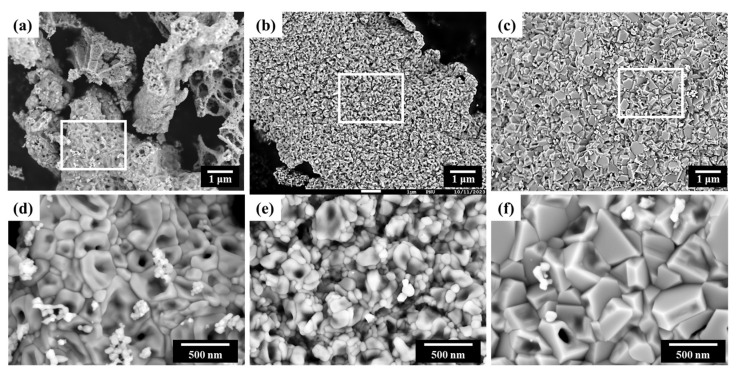
SEM images of Ni nanopowder with different F/O ratios of (**a**) 0.25, (**b**) 1.00, and (**c**) 2.00, with a magnification of 10,000×. The enlarged images of insets with a magnification of 50,000× are shown as (**d**), (**e**), (**f**), respectively.

**Figure 5 materials-16-07191-f005:**
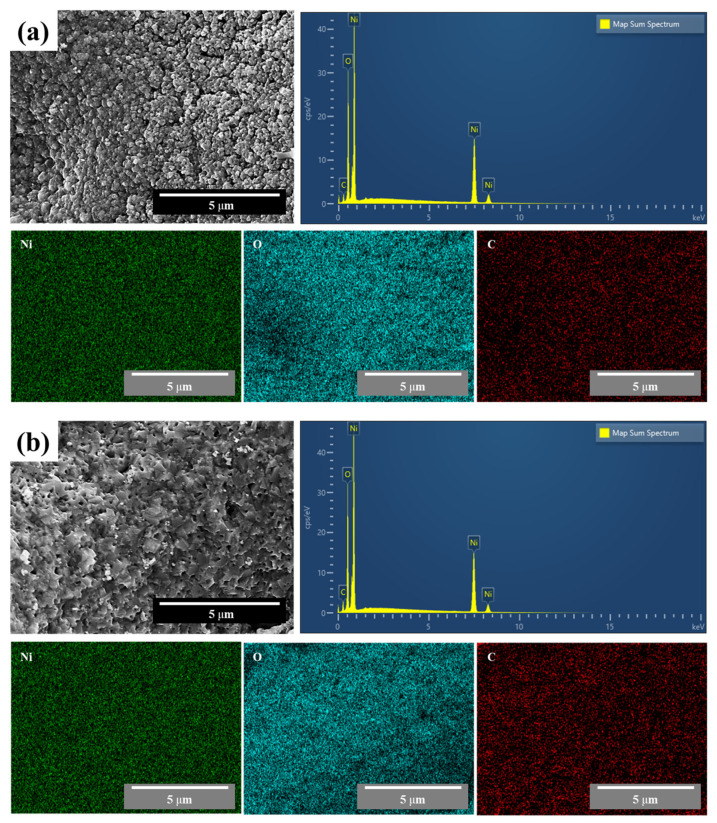
EDS results for synthesized NiO nanopowder with different F/O ratios of (**a**) 0.25 and (**b**) 2.00.

**Figure 6 materials-16-07191-f006:**
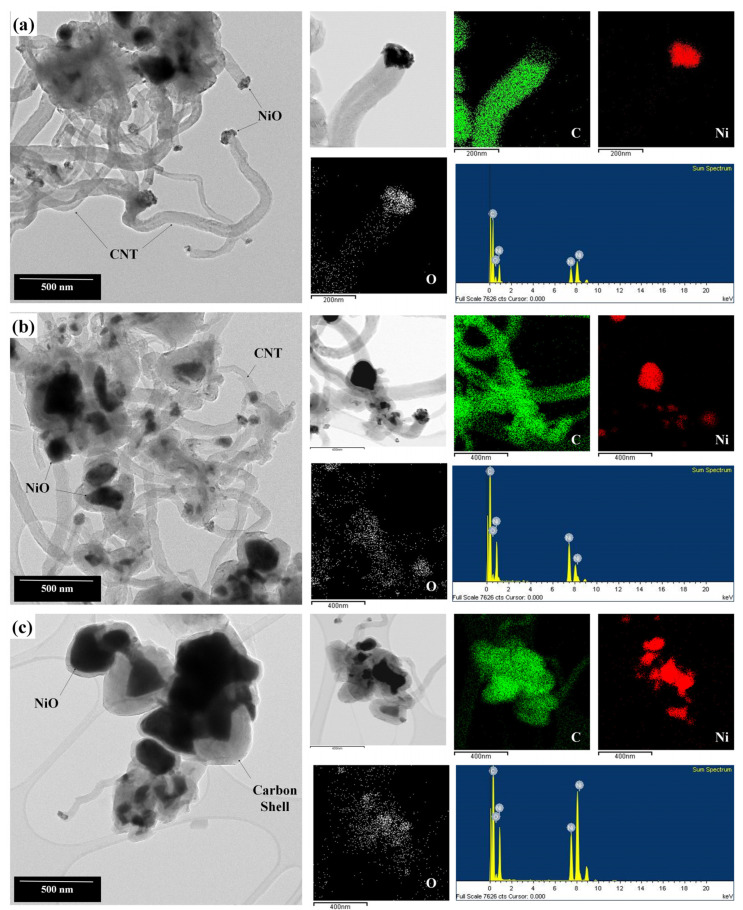
Morphology and composition analysis of CNTs grown from NiO nanocatalysts with different F/O ratios of (**a**) 0.25, (**b**) 1.00, and (**c**) 2.00.

**Figure 7 materials-16-07191-f007:**
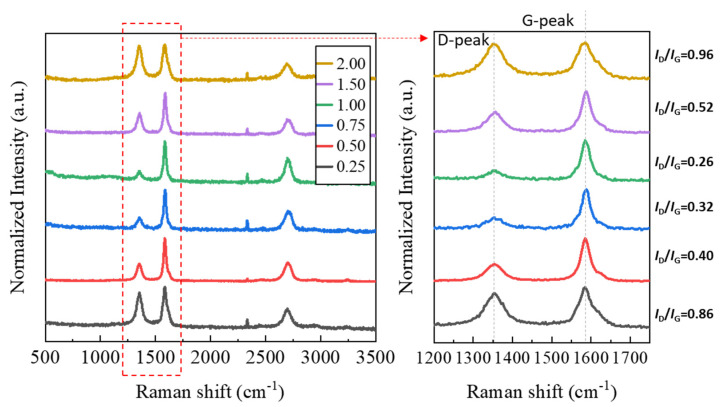
Raman spectrum analysis of combustion powders with different F/O ratios.

## Data Availability

The data and analyses in this study are available upon request from the corresponding authors.

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
