# Peer review of "Solution Combustion Synthesis of Ni-Based Nanocatalyst Using Ethylenediaminetetraacetic Acid and Nickel-Carbon Nanotube Growth Behavior"

_materials, 2023, doi:10.3390/ma16227191_

Round 1
Reviewer 1 Report
Comments and Suggestions for Authors
needs a little bit improvement
Author Response
The article by Kim et al. is designed to the influence of ethylenediaminetetraacetic acid (EDTA) content used as combustion fuel when fabricating nickel oxide (NiO) nano-catalysts by solution combustion synthesis, 15 as well as the growth behavior of CNTs using this catalyst. However, many unclear corrections should be made before considering it for publication.
Answer) Thank you for your kind comments. We revised the manuscript referring to your comments as follows.
Q 1) The authors must provide an expanded form of CNTs and CCVD in the abstract.
Answer) We revised abstract according to the comment.
Q 2) In line 27, the phrase ¨due to the presence of overgrown the catalyst particles¨ is unclear.
Answer) In line 27, we revised the sentence including the phrase as follows:
“As the F/O ratio increased, it was observed that the catalyst particles grew excessively beyond hundreds of nanometres, preventing further CNT growth and leading to a rapid termination of CNT growth.”
Q 3) In line 71, the phrase ¨serving as oxidizers in the SCS reaction. Nickel nitrate¨ is unclear.
Answer) We revised the sentence including the phrase as follows:
“Metal ions are introduced in the form of metal nitrates, metal sulfates, or similar com-pounds and are decomposed in the SCS reaction, with the metal ions becoming the nucleus from which the catalyst grows and the salts acting as oxidants [12].”
Q 4) In line 79, it would be more accurate to write ¨fuel-to-oxidizer ratio of NiO and EDTA¨ as ¨fuel-to-oxidizer ratio of EDTA and NiO¨.
Answer) We revised the sentence including the phrase as you comment “fuel-to-oxidizer ratio of EDTA and NiO”.
Q 5) In line 122, Ni2+ should be rewritten as Ni2+.
Answer) We revised Ni2+ as Ni2+.
Q 6) In Figure 3, the abbreviation for average should be the same in both images.
Answer) We revised the abbreviation of figures as same in both images as “avg.”.
Q 7) The heading ¨Carbon nanotube growth by CCVD¨ can be modified to ¨Carbon nanotube growth by CCVD approach¨.
Answer) As you comment, it would be more appropriate to use a word with the meaning of "method" alongside CCVD since it is just the name of a method. As the CCVD method is already widely accepted, we have changed the title of section 3.3 to use the word "method".

Reviewer 2 Report
Comments and Suggestions for Authors
Kim and coworkers present the influence of EDTA in fabricating nickel oxide catalyst for carbon nanotube growth. By tuning the ratio of the metal precursor and fuel EDTA, the proportion of Ni metal can be fine tuned, as well as the crystallite size. The carbon nanotube growth will also be influenced by using different NiO nano-catalyst to change the crystallinity. This work is well conducted and presented. After addressing the following minor issues, this work can be accepted.
1) In Figure 3 of the average crystallite size of NiO, it is better to add the error bar of the average size. The size measurement/method should also be mentioned.
2) What is the diameter of the CNT. Will the nanocatalyst influence the diameter/size of the CNT? This should be discussed besides of the crystalline.
Author Response
[Comments from Reviewer 2]
Kim and coworkers present the influence of EDTA in fabricating nickel oxide catalyst for carbon nanotube growth. By tuning the ratio of the metal precursor and fuel EDTA, the proportion of Ni metal can be fine tuned, as well as the crystallite size. The carbon nanotube growth will also be influenced by using different NiO nano-catalyst to change the crystallinity. This work is well conducted and presented. After addressing the following minor issues, this work can be accepted.
Answer)
Thank you for your kind comment. We revised the manuscript referring to your comments as follows.
Q 1)
You will see that a number of general and specific points are mentioned which necessitate extensive rewriting of the paper. You will see that a number of general and specific points are mentioned which necessitate extensive rewriting of the paper. In Figure 3 of the average crystallite size of NiO, it is better to add the error bar of the average size. The size measurement/method should also be mentioned.
Answer) We have included an error bar in figure 3 in accordance with your comment. The crystalline size was calculated using the Scherrer equation based on the information of the main peaks obtained from the XRD measurements as mentioned in the manuscript.
Q 2) What is the diameter of the CNT. Will the nanocatalyst influence the diameter/size of the CNT? This should be discussed besides of the crystalline.
Answer) The diameters of the grown CNTs were observed to be between 150 and 210 nm.
Several studies have shown that the morphology of the grown CNTs depends on the diameter/size of the catalyst particles [15, 26]. Single-walled CNTs can be expected to grow on particles as small as a few nm, multi-walled CNTs can be expected to grow on particles as large as tens to hundreds of nm (although there are differences in the literature, generally below 200-300 nm), and CNTs are expected to be difficult to grow beyond that. The growth behaviour of CNTs according to the size of the catalyst particles grown in this study is described in Section 3.3.

Reviewer 3 Report
Comments and Suggestions for Authors
This study focuses on analyzing the influence of fuel-to-oxidizer (F/O) ratios on catalyst morphologies and carbon nanotube (CNT) growth. The characterization of the Ni-based catalysts revealed that an F/O ratio of 1 resulted in the smallest particle size. Additionally, the authors found that the most favorable crystallinity of CNTs was achieved using an F/O ratio of 1. The mechanistic studies conducted by the authors led to several hypotheses that could potentially explain this phenomenon. This work represents an important discovery in highlighting the significant impact of F/O ratios on CNT growth, with the conclusion well supported by comprehensive characterization and mechanistic studies. In my opinion, this work holds significance in the field of materials science and should be of interest to the readers of this journal. Consequently, I recommend the publication of this work on Materials. However, there are a few concerns and suggestions that should be addressed before publication.
1. In line 71, the authors mentioned "Nickel nitrate" but did not complete the sentence properly.
2. Line 108, 'a positive Ø implies a feel-lean composition', but it should be the opposite.
3. In line 152, the authors made a reference to the phase analysis results in Figure 2 confirming a point. It is necessary for the authors to provide a more detailed explanation and elaboration on how these results further support their findings.
4. Figure 1c should be discussed in the manuscript.
Author Response
[Comments from Reviewer 3]
This study focuses on analyzing the influence of fuel-to-oxidizer (F/O) ratios on catalyst morphologies and carbon nanotube (CNT) growth. The characterization of the Ni-based catalysts revealed that an F/O ratio of 1 resulted in the smallest particle size. Additionally, the authors found that the most favorable crystallinity of CNTs was achieved using an F/O ratio of 1. The mechanistic studies conducted by the authors led to several hypotheses that could potentially explain this phenomenon. This work represents an important discovery in highlighting the significant impact of F/O ratios on CNT growth, with the conclusion well supported by comprehensive characterization and mechanistic studies. In my opinion, this work holds significance in the field of materials science and should be of interest to the readers of this journal. Consequently, I recommend the publication of this work on Materials. However, there are a few concerns and suggestions that should be addressed before publication.
Answer) Thank you for your kind comments. We revised the manuscript referring to your comments as follows.
Q 1) In line 71, the authors mentioned "Nickel nitrate" but did not complete the sentence properly.
Answer) We deleted the pointed words “Nickel nitrate”.
Q 2) Line 108, 'a positive Ø implies a feel-lean composition', but it should be the opposite.
Answer) Thank you for your comment. However, if the value of Ø in the above sentence is greater than 1, it indicates that the synthesis reaction cannot be completed using only Ni(NO3)2∙6H2O and EDTA as starting materials, and that additional O2 is needed, resulting in a fuel-lean state according to the pre-reaction calculation in Eq. (1). Similarly, if the value of Ø is less than 1, it shows a surplus of O2 resulting from the decomposition of the added EDTA, creating a fuel-rich state.
Q 3) In line 152, the authors made a reference to the phase analysis results in Figure 2 confirming a point. It is necessary for the authors to provide a more detailed explanation and elaboration on how these results further support their findings.
Answer) We revised the text related to phase analysis to explain it in more detail.
Q 4) Figure 1c should be discussed in the manuscript.
Answer) We've added the following paragraphs on Figure 1c in response to your comment.
“Thus, the production of nano-catalysts involved heat treatment at the temperatures of 260 °C, marking the complete termination of the SCS reaction, and 600 °C, which results in the decomposition of Ni(NO3)2∙6(H2O) forming NiO. It is illustrated in the heat treatment profile, as depicted in Figure 1.c.”

Reviewer 4 Report
Comments and Suggestions for Authors
Summary:
The authors presented the fabrication of NiO nanocatalyst using solution combustion synthesis. The results showed that the ratio between EDTA and the oxidizer has a significant impact on the ratio of NiO as well as the crystallite size within the final catalysts. Their catalytic performance in CNT growth were also experimentally evaluated.
Impact:
- The authors presented a novel approach to prepare NiO catalysts, investigated the synthetic procedures in detail and found that the F/O ratio can have a significant effect on the catalysts produced. This will be very helpful and instructive for researchers to do similar experiments in the future.
- The authors also evaluated the performance of catalysts, they found the best catalytic condition for CNT growth and that excess NiO could have a negative effect on the reaction. This can help researchers working in similar fields to gain better control of their experimental conditions.
- The results are well presented and supported by experimental data.
- This paper can be of great interest to readers of Materials, especially for those working on catalysts and nanomaterials.
This manuscript shows exciting findings, and the results are well presented. Based on my evaluations above, this manuscript shows I suggest accepting the manuscript as it is.
Author Response
[Comments from Reviewer 4]
The authors presented the fabrication of NiO nanocatalyst using solution combustion synthesis. The results showed that the ratio between EDTA and the oxidizer has a significant impact on the ratio of NiO as well as the crystallite size within the final catalysts. Their catalytic performance in CNT growth were also experimentally evaluated.
Impact:
- The authors presented a novel approach to prepare NiO catalysts, investigated the synthetic procedures in detail and found that the F/O ratio can have a significant effect on the catalysts produced. This will be very helpful and instructive for researchers to do similar experiments in the future.
- The authors also evaluated the performance of catalysts, they found the best catalytic condition for CNT growth and that excess NiO could have a negative effect on the reaction. This can help researchers working in similar fields to gain better control of their experimental conditions.
- The results are well presented and supported by experimental data.
- This paper can be of great interest to readers of Materials, especially for those working on catalysts and nanomaterials.
This manuscript shows exciting findings, and the results are well presented. Based on my evaluations above, this manuscript shows I suggest accepting the manuscript as it is.
Answer)
Thank you for your kind comment. We will continue our research in this area based on your comments.
